# Association between ischemic stroke and seropositive rheumatoid arthritis in Korea: A nationwide longitudinal cohort study

Dong Hyun Lee[1,2], Seung Hun Sheen[2], Dong-Geun Lee[1], Jae-Won Jang[1], Dong Chan Lee[3], Seung-Ho Shin[3], In-bo Han[2], Je Beom Hong[4], Hakyung Kim[5], Seil Sohn[2]*

1 Department of Neurosurgery, Spine Center, The Leon Wiltse Memorial Hospital, Suwon, Gyeonggi-do, South Korea, 2 Department of Neurosurgery, CHA University, CHA Bundang Medical Center, Seongnam, Gyeonggi-do, South Korea, 3 Department of Neurosurgery, Spine Center, The Leon Wiltse Memorial Hospital, Anyang, Gyeonggi-do, South Korea, 4 Kangbuk Samsung Hospital, Sungkyunkwan University College of Medicine, Seoul, Korea, 5 Department of Public Health, Genome & Health Big Data Branch, Graduate School of Public Health, Seoul National University, Seoul, Korea

* sisohn@cha.ac.kr

## Abstract

The purpose of this longitudinal follow-up study was to investigate the risk of ischemic stroke nationwide in patients with seropositive rheumatoid arthritis (RA) and controls who were matched in age and sex. Patient data were collected from the National Health Insurance Service (NHIS) Health Screening (HEALS) cohort. Using the International Classification of Diseases code M05 (seropositive RA), with a prescription of any disease-modifying anti-rheumatic drug (DMARD), RA was identified. A total of 2,765 patients and 13,825 control subjects were included in our study. The 12-year incidence of ischemic stroke in each group was calculated using the Kaplan–Meier method. The risk ratio of ischemic stroke was estimated using Cox proportional hazards regression. Sixty-four patients (2.31%) in the seropositive RA group and 512 (3.70%) in the control group experienced ischemic stroke (P < 0.001) during the follow-up period. The hazard ratio of ischemic stroke in the seropositive RA group was 1.32 (95% confidence interval (CI), 1.02–1.73) after adjusting for age and sex. The adjusted hazard ratio of ischemic stroke in the seropositive RA group was 1.40 (95% CI, 1.07–1.82) after adjusting for demographics and comorbid medical disorders. According to the subgroup analysis, the hazard ratios of ischemic stroke risks in the female and hypertensive subgroups were 1.44 (95% CI, 1.05–1.97) and 1.66 (95% CI, 1.16–2.38), respectively. In the non-diabetes and non-dyslipidemia subgroups, the corresponding hazard ratios of ischemic stroke were 1.47 (95% CI, 1.11–1.95) and 1.43 (95% CI, 1.07–1.91). Seropositive RA patients have an increased risk of ischemic stroke. In female, hypertension, non-diabetes, and non-dyslipidemia RA subgroups, even without the traditional risk factors for stroke (except for hypertension), increased the risk, which could be potentially attributed to RA.

**Data Availability Statement:** Data are available from the National Health Insurance Service (NHIS) Health Screening (HEALS) cohort (http://nhiss.

nhis.or.kr/) for researchers who meet the criteria for access to confidential data. While NHIS restricts the data from being shared publicly, interested researchers can apply for access to the data at the NHIS-HEALS web site (http://nhiss.nhis.or.kr/, tel: + 82-33-736-2430/ 2431). Upon approval, the data will be provided directly from NHIS-HEALS to the individual researchers. These processes are intended to get consent from all researchers for the compliance of ethical guidelines not to impede the data sharing. We did not have special access privileges to these data sets.

**Funding:** This study was funded by the Basic Science Research Program through the National Research Foundation of Korea (NRF) funded by the Ministry of Science, ICT and Future Planning (https://www.nrf.re.kr.), grant number NRF-2020R1F1A1069875 to SS. The funders had no role in study design, data collection and analysis, decision to publish, or preparation of the manuscript.

**Competing interests:** The authors have declared that no competing interests exist.

# Introduction

Rheumatoid arthritis (RA) is a chronic, systemic, immune-mediated inflammatory disorder that induces dysfunction [1] and premature mortality [2–4]. The long-term RA prognosis has greatly improved with the introduction of effective disease-modifying anti-rheumatic drugs (DMARDs); however, the overall RA burden is increasing, due to increases in chronic diseases and life expectancy worldwide [5].

RA is reported to be primarily associated with a decrease in median survival rate [6] that is due to cardiovascular complications. In particular, the frequency, clinical severity, and mortality of acute coronary syndromes are increased in RA patients [7–9]. With regard to ischemic stroke, the picture is less clear and the data is contradictory [10, 11]. A few studies reported a higher incidence of ischemic stroke among patients with RA [10, 12–14], whereas others have reported no link between the two factors [11, 15, 16]. Regarding mortality, some studies reported increased mortality rates, whereas others reported no differences [6, 17–21]. These contradictory results pertaining to ischemic stroke have been uncovered not only in RA but also in other inflammatory diseases, such as ankylosing spondylitis [22–24].

Moreover, previous studies have concentrated on the mortality and disability rates and the economic burden of RA in Korea [25]. However, an association between ischemic stroke and seropositive RA has not been reported for the country. Therefore, this nationwide longitudinal follow-up study aimed to clarify the risk of ischemic stroke in a seropositive RA population in Korea.

# Materials and methods

## Data source

We acquired the rights to use it from the Institutional Review Board (IRB No. 2020-01-011) of the Bundang Medical Center, CHA University. Since the data used in this study was anonymous secondary data released for research purposes, prior consent was not required. The Institutional Review Board (IRB No. 2020-01-011) of the Bundang Medical Center, CHA University provided approval for the study.

South Korea provides a single-payer health insurance system that is supervised and managed by the National Health Insurance Service (NHIS). All medical providers are required to submit patient claims to the NHIS, and the claimed amount is reimbursed after review. The NHIS provides general health examinations every 1 or 2 years for full-time and part-time workers as well as unemployed individuals aged 40 years or above. The collected patient data (e.g., demographic profiles, health insurance claims, national health examination results, disability certificates, and death certificates) are stored in the National Health Information Database (NHID). The NHIS compiled the NHIS-HEALS database representing a cohort that participates in national health examinations. The current study includes data obtained from the NHIS-HEALS cohort from 2004 to 2015.

The NHIS collects data on sociodemographic parameters (age, gender, average insurance premium, residential area, and presence of disability), clinical information (comorbidities, number of outpatient visits, and hospitalization records), and the results from a national health screening program.

## Establishment of the study cohort

The NHIS-HEALS cohort included 514,557 participants, approximately 10% of the Korean population from 2002 to 2003, aged between 40 and 79 and who had undergone national health screening. It was followed up for 12 years, until December of 2015 [26]. Rheumatoid

factor (RF) and anti-cyclic citrullinated peptide (CCP) antibody tests are often measured to diagnose rheumatoid arthritis (RA); however, the tests may not be repeated, which makes it difficult to extract the needed information correctly from electronic health record data. Considering these points, we used M5 (seropositive) diagnostic ICD-10 code based on these factors [27]. The study population included a seropositive RA group and a control group. The seropositive RA group consisted of subjects with a diagnosis of seropositive RA (any ICD-10 code starting with M05) with a prescription of any disease-modifying anti-rheumatic drug (DMARD) between January 1, 2004, and December 31, 2015 [25, 28]. Our study based on people who received conventional synthetic DMARD prescriptions, such as hydroxychloroquine, methotrexate, sulfasalazine and leflunomide. Biological DMARD medications were examined, whether adalimumab, etanercept, infliximab, or rituximab were prescribed. We extracted 9,120 seropositive RA subjects from a total of 514,557 patients in the NHIS-HEALS cohort. After excluding 1,384 subjects with preexisting seropositive RA, a total of 7,736 patients with newly diagnosed seropositive RA remained. After exclusion of those without a prescription for any DMARD, 2765 patients with RA were retained. After applying this selection criteria, 2765 patients with newly diagnosed RA were identified between 2004 and 2015 as our base cohort from the NHIS-HEALS cohort for analysis. Through 1:5 age- and sex-stratified matching (without replacement) using a greedy-match algorithm of the R package "MatchIT" [29, 30], 13,825 individuals were chosen as controls (Fig 1). Subjects in this study were followed up from the first occurrence of ischemic stroke until death or end of follow-up. The risk of ischemic stroke was assessed after adjusting for age, sex, and comorbidities, including hypertension, diabetes mellitus, and dyslipidemia. Information on existing comorbidities was extracted by searching all outpatient and inpatient medical records in the NHIS-HEALS cohort database.

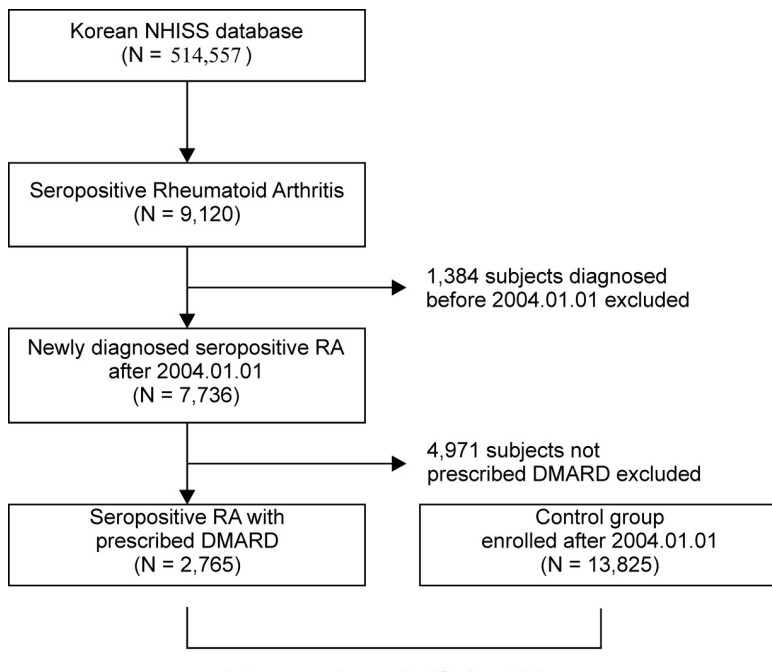

**Fig 1. Cohort creation flow diagram.** This study was a 12-year longitudinal cohort study established from the NHIS-HEALS cohort.

## Definitions of ischemic stroke and comorbidities

Among the seropositive RA and control group, ischemic stroke patients were those with ICD-10 codes I63 or I64; hospitalized for ischemic stroke, as confirmed by brain CT or MRI scans [22, 31]; and hospitalization ≥1. In the seropositive RA group, ischemic stroke patients were limited to the case of receiving the ICD-10 codes I63 or I64 after diagnosed with the M5. Comorbidities such as diabetes mellitus (E11–E14), hypertension (I10–I13, I15), and dyslipidemia (E78) were defined by ICD-10 codes with additional information described previously [32–34] (Table 1).

## Statistical analysis

To compare demographic characteristics and complications between the seropositive RA group and the control group, chi-squared tests and student's t-tests were used to analyze differences between the means. The Kaplan–Meier method was used to obtain the likelihood of survival without ischemic stroke in each group. To estimate the effects of seropositive RA on the later occurrence of each event, multivariate analysis was performed using a Cox proportional hazards regression model. Specifically, two Cox proportional hazards regression models were used. We used two corrected models, where Model 1 was adjusted for age and sex and Model 2 was adjusted for age, sex, and other accompanying diseases (Table 2). Subgroup analyses were also performed using Cox proportional hazards regression models to estimate the effects of seropositive RA on the risk of each event (Table 3). Data analysis was performed using R software (version 3.3.3).

## Results

### Characteristics of the seropositive RA and control groups

The newly diagnosed seropositive RA patients (2,765) who were prescribed with DMARD during the study period were mostly females (73.4%), with a mean age of 54 ± 8.7 years. Significant differences were found between the two groups in the prevalence of diabetes, hypertension, dyslipidemia, and ischemic stroke. Diabetes prevalence was higher in the control group (10.10%) than in the RA group (6.26%) (p <0.001). Additionally, hypertension prevalence was higher in the RA group (30.13%) than in the control group (26.44%) (p <0.001). Dyslipidemia prevalence was higher in the control group (17.84%) than in the RA group (15.55%)

**Table 1. Definitions of ischemic stroke and comorbidities.**

| Diagnosis | ICD-10 codes | ICD-10 codes and definitions |
|---|---|---|
| Hypertension[a] | I10-I13, I15 | At least one claim per year for the prescription of |
| | | Anti-hypertensive medication (thiazide, loop diuretics, aldosterone antagonist, alpha-/beta-blocker, calcium-channel blocker, angiotensin-converting enzyme inhibitor, angiotensin II receptor blocker) under ICD-10 code I10-13 or I15. |
| Type 2 DM[b] | E11-E14 | At least one claim per year for the prescription of |
| | | Anti-diabetic medication (sulfonylureas, metformin, meglitinides, thiazolidinediones, dipeptidyl peptidase-4 inhibitors, α-glucosidase inhibitors, and insulin) under ICD-10 code E11-14. |
| Dyslipidemia[b] | E78 | At least one claim per year for the prescription of anti-dyslipidemic medication under ICD10 code E78 |
| Ischemic stroke | I63, I64 | Hospitalized ≥ 1 day for ischemic stroke, as confirmed by brain CT or MRI scans |

[a] ≥ 1 diagnosis during hospitalization, or ≥ 2 diagnoses at outpatient clinic, in the previous 1 year.

[b] ≥ 1 diagnosis during hospitalization or at outpatient clinic, in the previous 1 year.

**Table 2. Adjusted hazard ratio for ischemic stroke in the seropositive RA and control groups.**

| Disease | Group | N | Event | Duration (days) | Incidence rate (1000 person-year) | Hazard Ratio | | | |
|---|---|---|---|---|---|---|---|---|---|
| | | | | | | (95% CI) | | | |
| | | | | | | Model 1 | P values | Model 2 | P values |
| **Stroke** | Control | 13,825 | 512 | 55273331 | 3.38 | 1 | 0.039 | 1 | 0.014 |
| | RA | 2,765 | 64 | 5590704 | 4.18 | 1.32 (1.02–.73) | | 1.40 (1.07–1.82) | |

Abbreviations: CI, confidence interval; RA, rheumatoid arthritis.

Model 1: Adjusted for age and sex.

Model 2: Adjusted for age, sex, income, diabetes, hypertension, and dyslipidemia.

(p = 0.004). Ischemic stroke prevalence was higher in the control group (3.70%) than in the RA group (2.31%) (p <0.001) (Table 4).

## Ischemic stroke in the seropositive RA and control groups

The incidence rate of ischemic stroke was higher in the seropositive RA group (4.18) than in the control group (3.38) per 1000 person-year. The hazard ratios of ischemic stroke in the seropositive RA group were 1.32 (95% CI, 1.02–1.73) and 1.40 (95% CI, 1.07–1.82) according to models 1 and 2, respectively (Table 2). The cumulative incidence of ischemic stroke was significantly higher in the seropositive RA group than in the control group (p = 0.014, Fig 2).

## Subgroup analysis of ischemic stroke incidence rate

The incidence of ischemic stroke did not differ significantly between the two groups among male subjects. However, when we limited our analysis to the female population, we observed that the ischemic stroke incidence rate was significantly greater in the seropositive RA group. The hazard ratio of ischemic stroke among female patients in the seropositive RA group was 1.44 (95% CI, 1.05–1.97) (Table 3). Similarly, we observed an increased incidence rate of ischemic stroke in the seropositive RA population when compared to controls in individuals without diabetes and dyslipidemia. The hazard ratio of ischemic stroke in the seropositive RA

**Table 3. Subgroup analyses between the seropositive RA and control groups.**

| Variables | | Seropositive RA | | Control | | Hazard Ratio | P-values |
|---|---|---|---|---|---|---|---|
| | | N | Incidence Rate (1000 person-year) | N | Incidence Rate (1000 person-year) | (95% CI) | for difference |
| **Sex** | Male | 18 | 4.80 | 178 | 4.65 | 1.12 (0.682–1.835) | P<0.001 |
| | Female | 46 | 4.07 | 334 | 2.95 | 1.44 (1.048–1.971) | |
| **Age** | <65 | 38 | 2.81 | 333 | 2.49 | 1.30 (0.922–1.832) | P<0.001 |
| | ≥65 | 26 | 14.33 | 179 | 1.02 | 1.37 (0.900–2.082) | |
| **Diabetes** | N | 7 | 0.49 | 100 | 0.73 | 1.47 (1.106–1.952) | P<0.001 |
| | Y | 57 | 59.60 | 412 | 28.14 | 0.98 (0.452–2.124) | |
| **Hypertension** | N | 29 | 2.72 | 240 | 2.48 | 1.14 (0.769–1.687) | P<0.001 |
| | Y | 35 | 7.54 | 272 | 4.99 | 1.66 (1.156–2.378) | |
| **Dyslipidemia** | N | 55 | 4.27 | 403 | 3.24 | 1.43 (1.073–1.912) | P = 0.008 |
| | Y | 9 | 3.69 | 109 | 4.02 | 0.92 (0.462–1.833) | |

Abbreviations: RA, rheumatoid arthritis; CI, confidence interval.

**Table 4. Characteristics in the seropositive RA and control groups.**

| Variables | RA (*n* = 2,765) | Control (*n* = 13,825) | P |
|---|---|---|---|
| Male | 735 (26.6%) | 3675 (26.6%) | |
| Age | 54 ± 8.7 | 54 ± 8.7 | |
| Diabetes | 173 (6.26%) | 1396 (10.10%) | <0.001 |
| Hypertension | 833 (30.13%) | 5038 (26.44%) | <0.001 |
| Dyslipidemia | 430 (15.55%) | 2466 (17.84%) | 0.004 |
| Ischemic stroke | 64 (2.31%) | 512 (3.70%) | <0.001 |

Bold p-values indicate statistical significance.

Abbreviation: RA, rheumatoid arthritis.

group without diabetes was 1.47 (1.11–1.95). The hazard ratio of ischemic stroke in the seropositive RA group without dyslipidemia was 1.43 (95% CI, 1.07–1.91). In the seropositive RA group with hypertension, the ischemic stroke incidence rate was significantly greater. The hazard ratio of ischemic stroke in the seropositive hypertension RA group was 1.66 (95% CI, 1.16–2.38). There were no significant differences in stroke incidence in the other subgroups.

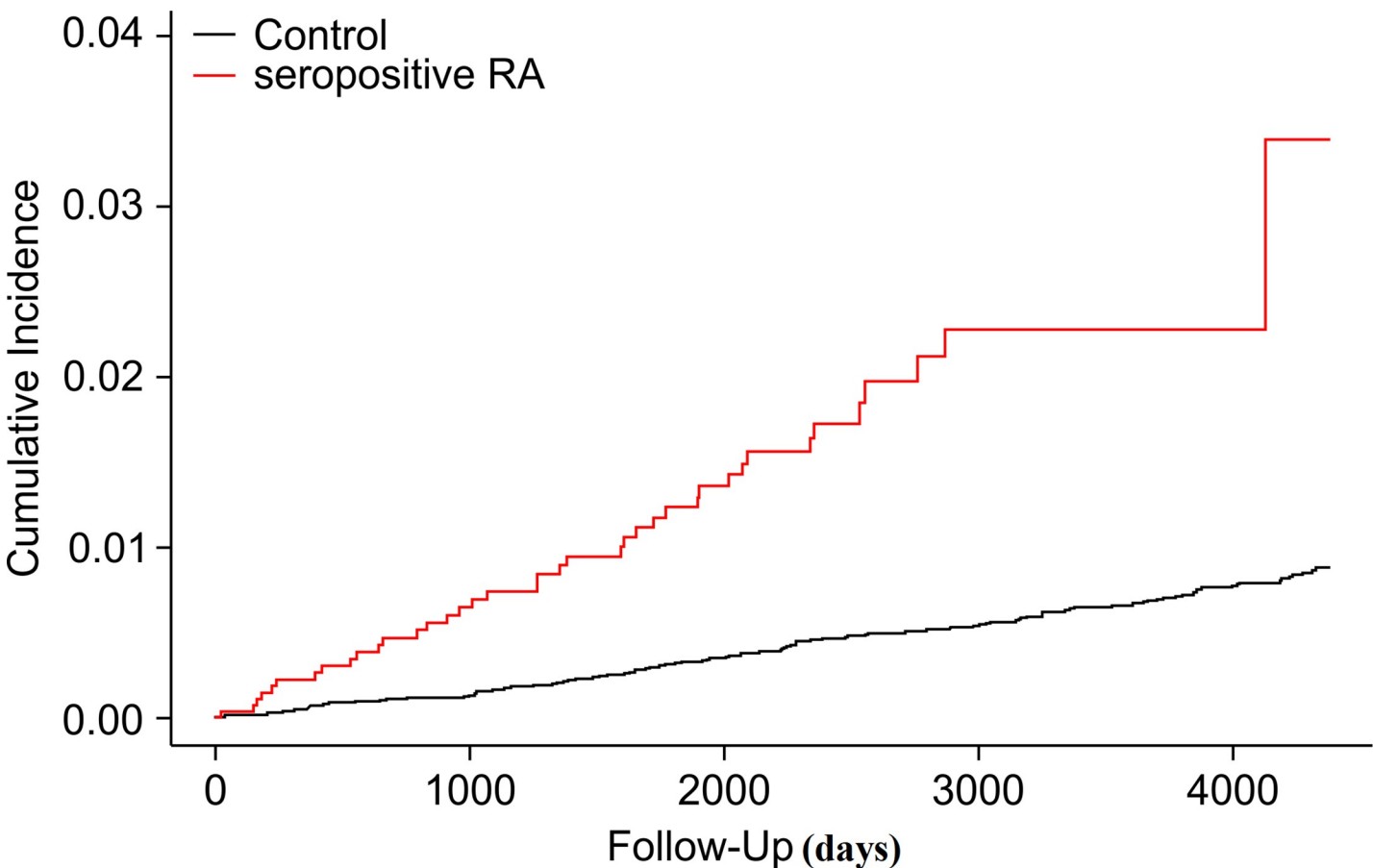

**Fig 2. Comparison of the cumulative incidence of ischemic stroke in the seropositive RA and control groups.** The Kaplan–Meier curves with cumulative hazards of ischemic stroke were compared between seropositive RA and control groups (*p* = 0.014).

## Discussion

We analyzed the association between seropositive RA and ischemic stroke in Korea. Our nationwide longitudinal cohort study showed an increased risk of ischemic stroke in patients with seropositive RA. Furthermore, we conducted a subgroup analysis.

Solomon et al. [11] reported that the adjusted relative risk of myocardial infarction, not ischemic stroke, was higher in women with RA than that in women without RA. Similarly, Turesson et al. [15] reported that there was a trend toward an increased incidence of first episodes of ischemic stroke in RA cohort; however, the difference was not statistically significant. In a nationwide Swedish rheumatology cohort, the hazard ratio for ischemic stroke was 1.29 (95% CI, 1.18–1.41) in RA patients [10]. These findings are similar to ours; in our study, the age- and sex-adjusted (model 1) hazard ratio of ischemic stroke in the seropositive RA group was 1.32 (95% CI, 1.02–1.73). In model 2 (which adjusted for comorbidities in addition to age and sex), the incidence rate of ischemic stroke was 1.40 (95% CI, 1.07–1.82).

Several studies have found no association between ischemic stroke and RA [11, 35, 36]. These results suggest a lack of power due to an insufficient number of ischemic stroke events or a weaker association between RA and ischemic stroke. Our nationwide longitudinal cohort study helped to address some of the above uncertainties.

According to the subgroup analysis, in RA patients with hypertension, the hazard ratios of ischemic stroke risks was 1.66 (95% CI, 1.16–2.38). Our results suggest that the risk of ischemic stroke increases in hypertensive RA patients. Hypertension may have caused damaging effects in the blood vessels of these patients. Accelerated atherosclerosis caused by inflammation may have increased arterial stiffness [37], which may explain the evolution of ischemic stroke risks observed in our cohort.

Our results showed that the prevalence of diabetes and hyprelipidemia was lower in the RA group than in the control group. The prevalence of diabetes was lower in the RA group (6.26%) than in the control group (10.10%) (p <0.001). The prevalence of hyperlipidemia was lower in the RA group (15.55%) than in the control group (17.84%) (p = 0.004). The above-mentioned results are similar to those reported in other studies [38, 39]. One study [38] reported that diabetes mellitus tended to be less common in the RA group than in controls, and that the RA patients (4.6–6.3 mmol/L) had significantly lower total serum cholesterol than that in controls (5.4–7.0 mmol/L) (P < 0.001). A previous study [39] supported our results by reporting a lesser prevalence of hyperlipidemia in the RA group. In some studies [40, 41], diabetes showed a higher prevalence in the RA group than in the control group, but there was no significant difference in the prevalence of hyperlipidemia between the two groups. There results demonstrates controversy in the prevalence of diabetes and hyperlipidemia among patients with RA compared to controls. Therefore, further studies may be warranted.

Smoking is a well-established risk factor for RA [42]. Moreover, smoking and inflammatory activity have been demonstrated to increase fibrinogen levels, and this increase is a known risk factor for thromboembolic events such as ischemic stroke [43]. Similar to smoking, hypercoagulation and thrombosis caused by the inflammatory reaction in RA, which leads to embolization, are described as factors that increase the risk of ischemic stroke in RA.

Atrial fibrillation is a crucial and remediable risk factor for stroke [44]. The incidence of atrial fibrillation is known to increase due to several more common factors in RA patients, such as smoking, ischemic heart disease, and heart failure [45–48]; Nonetheless, the relationship between atrial fibrillation and ischemic stroke is uncertain in RA patients [49]. However, research has shown that inflammatory reactions increase atrial fibrillation [50], and the increased risk of atrial fibrillation is expected to play an important role in the development of ischemic stroke in RA patients. To better understand the relevance of risk factors associated

with ischemic stroke and RA, researchers should address the degree of involvement of risk factors.

There were several limitations to this study. Due to the limitations of the NHIS database, information was lacking with regard to life-driven factors such as smoking or alcohol consumption, physical activity, eating patterns, and obesity. Furthermore, we were unable to provide information on RA clinical severity, such as disease activity score or radiographic findings. In addition, due to the insidious nature of certain vascular risk factors, such as diabetes, hypertension, and dyslipidemia, the NHIS-HEALS cohort data may be underdiagnosed. For this reason, the prevalence of vascular risk factors may have been less pronounced in both the seropositive RA and control groups. Nevertheless, to the best of our knowledge, this is the first study conducted in Korea that evaluates the association between ischemic stroke and seropositive RA in Korean patients. Our study is a nationwide, longitudinal study which included a large national data set based on health check-ups, with a follow-up that provided unified data to overcome the limitations of existing research results that used small amounts of data [38–41]. A large number of RA patients and their age-sex matched controls in a ratio of 1:5 also helped overcome the limitations of very low ischemic stroke incidence.

## Conclusion

Our nationwide population-based study suggests an increased risk of ischemic stroke in patients with seropositive RA in the Korean population. Particularly, this study suggests screening or intervention efforts to improve outcomes in female, hypertension, non-diabetes, and non-dyslipidemia RA patients.

## Author Contributions

**Conceptualization:** Dong Hyun Lee, Seil Sohn.

**Data curation:** Dong Hyun Lee, Seil Sohn.

**Formal analysis:** Dong Hyun Lee, In-bo Han, Je Beom Hong.

**Funding acquisition:** Dong Hyun Lee, In-bo Han, Je Beom Hong, Seil Sohn.

**Investigation:** Dong Hyun Lee, In-bo Han, Je Beom Hong, Seil Sohn.

**Methodology:** Dong Hyun Lee, Dong Chan Lee, Hakyung Kim, Seil Sohn.

**Project administration:** Dong Hyun Lee, Dong Chan Lee, Hakyung Kim, Seil Sohn.

**Resources:** Dong Hyun Lee, Seung Hun Sheen, Dong Chan Lee, Hakyung Kim, Seil Sohn.

**Software:** Dong Hyun Lee, Seung Hun Sheen, Dong-Geun Lee, Seung-Ho Shin.

**Supervision:** Dong Hyun Lee, Seung Hun Sheen, Jae-Won Jang, Seung-Ho Shin, Seil Sohn.

**Validation:** Dong Hyun Lee, Seung Hun Sheen, Seung-Ho Shin, Seil Sohn.

**Visualization:** Dong Hyun Lee, Seung Hun Sheen, Dong-Geun Lee.

**Writing – original draft:** Dong Hyun Lee.

**Writing – review & editing:** Dong Hyun Lee, Jae-Won Jang.

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
