## [Decision Letter · Decision Letter 0]

8 Dec 2020

PONE-D-20-30296

Association between ischemic stroke and seropositive rheumatoid arthritis in Korea: A nationwide longitudinal cohort study

PLOS ONE

Dear Dr. Sohn,

Thank you for submitting your manuscript to PLOS ONE. After careful consideration, we feel that it has merit but does not fully meet PLOS ONE’s publication criteria as it currently stands. Therefore, we invite you to submit a revised version of the manuscript that addresses the points raised during the review process.

The topic is very interesting, and the manuscript is well written, but several major factors should be acknowledged before considering this article to be published:

-** METHODS:**

**a) Clear inclusion and exclusion criteria should be written to understand both populations.**

**b) In terms of specific stroke risk, basic data from this condition should be provided, to understand if certain covariates could affect mortality and functional outcomes: severity of stroke cases (according to NIHSS), etiology of ischemic stroke (according to ASCOD, TOAST or any other classification criteria), type of acute therapy (IV thrombolysis, thrombectomy, both, or usual medical care), admission to stroke unit, general or rheumatology ward. Please clarifiy if this data is available, as I understand you adquire data from the NHID.**

**c) Clear definition on each risk factor should be added (i.e. hypertension)**

**RESULTS**

**a) P values are missing, on each one of the tables (HR). When adding statistical analysis for association, you should provide this values. **

**b) Kaplan Meier Curve should be corrected: Log-rank p value should be included, On the X axis, the authors should clarify if they are using person-year or years. Number at risk table should be included for each group, below X axis on each point of the analysis.**

**c) When including data from severity, type of stroke, and acute therapy I suggest to run again your analysis, to evaluate if the previous findings still remain in terms of covariates.**

We look forward to receiving your revised manuscript.

Kind regards,

Miguel A. Barboza, MD, MSc

Academic Editor

PLOS ONE

Journal Requirements:

2. Please include the date(s) on which you accessed the databases or records to obtain the data used in your study.

3. Thank you for submitting your revised manuscript and for improving the text overlap instances. There are still a few instances of text overlap with the following publications that need to be addressed:

https://www.jocn-journal.com/article/S0967-5868(20)31310-2/fulltext

https://ard.bmj.com/content/72/4/541

Please revise the manuscript to rephrase the duplicated text, cite your sources, and provide details as to how the current manuscript advances on previous work. Please note that further consideration is dependent on the submission of a manuscript that addresses these concerns about the overlap in text with published work.

We will carefully review your manuscript upon resubmission, so please ensure that your revision is thorough

4. We noticed you have some minor occurrence of overlapping text with the following previous publications, which needs to be addressed:

https://www.jocn-journal.com/article/S0967-5868(20)31310-2/fulltext

https://link.springer.com/article/10.1007/s00701-018-3499-7

In your revision ensure you cite all your sources (including your own works), and quote or rephrase any duplicated text outside the methods section. Further consideration is dependent on these concerns being addressed.

Reviewers' comments:

Reviewer's Responses to Questions

**Comments to the Author**

1. Is the manuscript technically sound, and do the data support the conclusions?

Reviewer #1: Yes

Reviewer #2: Yes

2. Has the statistical analysis been performed appropriately and rigorously? 

Reviewer #1: Yes

Reviewer #2: Yes

3. Have the authors made all data underlying the findings in their manuscript fully available?

Reviewer #1: Yes

Reviewer #2: Yes

4. Is the manuscript presented in an intelligible fashion and written in standard English?

Reviewer #1: Yes

Reviewer #2: Yes

5. Review Comments to the Author

Editor:

-** METHODS:**

**a) Clear inclusion and exclusion criteria should be written to understand both populations.**

**b) In terms of specific stroke risk, basic data from this condition should be provided, to understand if certain covariates could affect mortality and functional outcomes: severity of stroke cases (according to NIHSS), etiology of ischemic stroke (according to ASCOD, TOAST or any other classification criteria), type of acute therapy (IV thrombolysis, thrombectomy, both, or usual medical care), admission to stroke unit, general or rheumatology ward. Please clarifiy if this data is available, as I understand you adquire data from the NHID.**

**c) Clear definition on each risk factor should be added (i.e. hypertension)**

**RESULTS**

**a) P values are missing, on each one of the tables (HR). When adding statistical analysis for association, you should provide this values. **

**b) Kaplan Meier Curve should be corrected: Log-rank p value should be included, On the X axis, the authors should clarify if they are using person-year or years. Number at risk table should be included for each group, below X axis on each point of the analysis.**

**c) When including data from severity, type of stroke, and acute therapy I suggest to run again your analysis, to evaluate if the previous findings still remain in terms of covariates.**

**d) Finally, the previous changes should be considered in terms of the Discussion section, if significant changes are found.**

Reviewer #1: In the introduction part you have to explain better the definition of rheumatoid arthritis in which it doesn’t describe the dysfunction produced by it.

It is important the mention the definition of seropositive rheumatoid arthritis and what kind of antibodies were used to classify the population.

In the study design and subjects would be better if you explain what kind of DMARD was prescribed (biological or synthetic)

Reviewer #2: This work approaches ischemic stroke in RA patients, a relevant complication considering the increased cardiovascular risk. It describes the incidence of ischemic stroke in Korean RA patients, obtaining data from a national database. Although it draws some interesting conclusions regarding stroke incidence and its associations, there are important aspects to be addressed.

Major points

1. I consider the most concerning point is that there are contradictory statements regarding the main objective of the study (incidence of stroke in RA patients). In lines 136 and 137, the statement is: “The incidence of ischemic stroke was lower in the seropositive RA group (2.31%, 64 patients) than in the control group (3.70%, 512 patients)”; whereas in lines 142 and 143, it is: “The incidence of ischemic stroke was significantly higher in the seropositive RA group than in the control group (p =0.014, Fig 2).” Did the authors mean “cumulative incidence” in the second statement? If they were referring to person-years, they must clarify and include that information in the text, and not only in the figure.

2. Authors should clarify why they included only patients with newly diagnosed RA, and not with prevalent RA. This could lead to bias and to a lower risk of ischemic stroke, since patients with a longer disease duration could have an even higher risk than those who were newly diagnosed in the established time period.

3. Authors should clarify whether the ischemic stroke was diagnosed after RA. Although that seems intuitive, it is not included in the Methods section.

Minor points:

1. Line 85: “Accessed the databases number of this paper is NHIS-2019-2-194.” This statement does not make sense.

2. Line 97: There is a ”27” which seems out of place.

3. Lines 134 and 135: although the authors mention that there were significant differences in the prevalence of diabetes, hypertension, dyslipidemia and stroke, they do not clarify which were more prevalent in the RA patients and which in the control group. Even though that information can be found on Table 3, since there are differences in each condition, I consider it should be added to the text.

4. Line 192: Is it atrial fibrosis or atrial fibrillation? Authors should also clarify why would that explain a higher risk of stroke in RA patients

5. Line 203: What do you mean by “radiation grade”?

6. Because of the nature of the study, we do not know the smoking status of the population. Since smoking is a risk factor for both RA and stroke, this should be discussed as a limitation.

7. The different prevalence of diabetes and dyslipidemia between both groups should be discussed.

6. PLOS authors have the option to publish the peer review history of their article (what does this mean?). If published, this will include your full peer review and any attached files.

Reviewer #1: No

Reviewer #2: No

---

## [Author Response · Author response to Decision Letter 0]

2 Feb 2021

Dear Editor: 

I wish to re-submit the manuscript titled “Association between ischemic stroke and seropositive rheumatoid arthritis in Korea: A nationwide longitudinal cohort study.” The manuscript ID is PONE-D-20-30296R1

The manuscript has been rechecked and the necessary changes have been made in accordance with the reviewers’ suggestions. The responses to all comments have been prepared and attached. 

We thank you and the reviewers for your thoughtful suggestions. The manuscript has benefited from these insightful suggestions. I look forward to working with you and the reviewers to move this manuscript closer to publication in the PLOS ONE.

Thank you for your consideration. I look forward to hearing from you.

---

## [Decision Letter · Decision Letter 1]

25 Feb 2021

PONE-D-20-30296R1

Association between ischemic stroke and seropositive rheumatoid arthritis in Korea: A nationwide longitudinal cohort study

PLOS ONE

Dear Dr. Sohn,

Thank you for submitting your manuscript to PLOS ONE. After careful consideration, we feel that it has merit but does not fully meet PLOS ONE’s publication criteria as it currently stands. Therefore, we invite you to submit a revised version of the manuscript that addresses the points raised during the review process.

We look forward to receiving your revised manuscript.

Kind regards,

Miguel A. Barboza, MD, MSc

Academic Editor

PLOS ONE

Reviewers' comments:

Reviewer's Responses to Questions

**Comments to the Author**

1. If the authors have adequately addressed your comments raised in a previous round of review and you feel that this manuscript is now acceptable for publication, you may indicate that here to bypass the “Comments to the Author” section, enter your conflict of interest statement in the “Confidential to Editor” section, and submit your "Accept" recommendation.

Reviewer #2: All comments have been addressed

2. Is the manuscript technically sound, and do the data support the conclusions?

Reviewer #2: Yes

3. Has the statistical analysis been performed appropriately and rigorously? 

Reviewer #2: Yes

4. Have the authors made all data underlying the findings in their manuscript fully available?

Reviewer #2: Yes

5. Is the manuscript presented in an intelligible fashion and written in standard English?

Reviewer #2: Yes

6. Review Comments to the Author

Reviewer #2: The authors addressed my previous comments correctly. However, there are still some aspects I consider should be corrected:

- There are several grammatical errors throughout the manuscript, they should be corrected.

- Abstract:

The following statement is not clear: “Patients with seropositive rheumatoid arthritis (RA) who were matched in age and sex.”

Authors should state that they studied both RA patients and controls, who were age- and sex-matched.

I consider the abstract conclusion is not accurate. “Due to the high risk of ischemic stroke, close observation is especially necessary in female, hypertension, non-diabetes, and non-dyslipidemia RA subgroups”.

Whereas RA patients who were female, hypertense and without diabetes or hypertension had a higher risk of strike than their non-RA counterparts, it does not mean that they were the group with the highest stroke risk. It should be interpreted that these patients, even without the traditional risk factors for stroke (except for hypertension), had a higher risk which could be potentially attributed to RA.

Introduction

This sentence does not make sense: “Unlike debilitating joint effects, RA is reported to be primarily associated with a decrease in median survival rate”. Please rephrase.

Line 84 and 85: Change “who” for “that” and delete the “]” sign.

Line 95: it should be “citrullinated”

Line 122: Is it hospitalization >1 day?

Results

Are incidence rates in Table 3 correct? In people with RA over 65 years of age, it is 14.33 vs 1.02 in controls, with no significant difference. Also, it is mentioned that patients with RA without diabetes had a higher risk than controls, and the incidence rate shown in the Table is lower on those patients (0.49 vs 0.73)

The p value for hypertension in Table 3 should be in patients with hypertension, not without

I would suggest deleting this statement “The hazard ratio of ischemic stroke among male patients in the seropositive RA group was 1.12 (95% CI, 0.68–1.84)”, since it was not significant and is shown in the Table.

Discussion

The discussion is somewhat difficult to follow; it lacks cohesion and fails to give potential explanations and/or show the relevance of some of the study findings.

As I mentioned in my previous comments, the different prevalence of diabetes and dyslipidemia between both groups should be added to the discussion (not just mentioned in the Results section).

The paragraph in lines 232 to 236 is not clear. The differential outcomes of myocardial infarction and stroke and then the different mechanisms leading to both of them do not appear to be related.

7. PLOS authors have the option to publish the peer review history of their article (what does this mean?). If published, this will include your full peer review and any attached files.

Reviewer #2: No

---

## [Author Response · Author response to Decision Letter 1]

10 Apr 2021

Response to Reviewers

Title of the Manuscript: 

Association between ischemic stroke and seropositive rheumatoid arthritis in Korea: A nationwide longitudinal cohort study

Manuscript Number: PONE-D-20-30296

Reviewer #2: The authors addressed my previous comments correctly. However, there are still some aspects I consider should be corrected:

Q: There are several grammatical errors throughout the manuscript, they should be corrected.

Author’s Response:

I appreciate your valuable comment and I understand your concern. Following your advice, we have reviewed and corrected grammatical errors throughout the manuscript. Our original manuscript was edited by a native speaker of English. However, after your comment, our revised manuscript was edited again by another native speaker of English who also studied biology. Please refer to the certificate attached herewith.

Abstract:

Q: The following statement is not clear: “Patients with seropositive rheumatoid arthritis (RA) who were matched in age and sex.”

Authors should state that they studied both RA patients and controls, who were age- and sex-matched.

Author’s Response:

I appreciate and agree with your comment. We have changed the sentence according to your comment as follows. 

Page 2, line 27

The purpose of this nationwide longitudinal follow-up study was to investigate the risk of ischemic stroke nationwide in patients with seropositive rheumatoid arthritis (RA) and controls who were age- and sex- matched.

Q: I consider the abstract conclusion is not accurate. “Due to the high risk of ischemic stroke, close observation is especially necessary in female, hypertension, non-diabetes, and non-dyslipidemia RA subgroups”.

Whereas RA patients who were female, hypertense and without diabetes or hypertension had a higher risk of strike than their non-RA counterparts, it does not mean that they were the group with the highest stroke risk. It should be interpreted that these patients, even without the traditional risk factors for stroke (except for hypertension), had a higher risk which could be potentially attributed to RA.

Author’s Response:

I appreciate your valuable comments and agree with you. Thanks to your comment, we could revise abstract conclusion. We've revised the abstract conclusion as follows.

Page 2, line 45

RA patients in female, hypertension, non-diabetes, and non-dyslipidemia subgroups, even without the traditional risk factors for stroke (except for hypertension), had a higher risk of ischemic stroke, which could be potentially attributed to RA.

Introduction

Q: This sentence does not make sense: “Unlike debilitating joint effects, RA is reported to be primarily associated with a decrease in median survival rate”. Please rephrase.

Author’s Response:

I appreciate and agree that the sentence should be rephrased. We modified the sentence as follows.

Page 3 line 56

RA is reported to be associated with a decrease in median survival rate that is primarily due to cardiovascular complications.

Q: Line 84 and 85: Change “who” for “that” and delete the “]” sign.

Author’s Response:

I appreciate your comment and agree with you. We modified the sentence as follows.

Page 4 line 84

The NHIS compiled the NHIS-HEALS database representing a cohort that participates in national health examinations.

Q: Line 95: it should be “citrullinated”

Author’s Response:

I appreciate your very much. Thanks to your comment, we corrected the sentence as follows. 

Page 4 line 95

Rheumatoid factor (RF) and anti-cyclic citrullinated peptide (CCP) antibody tests are often measured to diagnose rheumatoid arthritis (RA);

Q: Line 122: Is it hospitalization >1 day?

Author’s Response:

I appreciate your comment. The hospitalization is for 1 day or more than 1 day. It should be “hospitalization ≥1”. We revised it as follows. 

Page 4 line 95

Among the seropositive RA and control group, ischemic stroke patients were those with ICD-10 codes I63 or I64; hospitalized for ischemic stroke, as confirmed by brain CT or MRI scans [22,30]; and hospitalization ≥1.

Results

Q: Are incidence rates in Table 3 correct? In people with RA over 65 years of age, it is 14.33 vs 1.02 in controls, with no significant difference. Also, it is mentioned that patients with RA without diabetes had a higher risk than controls, and the incidence rate shown in the Table is lower on those patients (0.49 vs 0.73)

Author’s Response:

I appreciate your comment and understand your concerns. In table 3, subgroup analyses were performed using Cox proportional hazards regression models to estimate the effects of seropositive RA on the risk of each event. Confidence interval of hazard ratio was affected by follow-up period. In over 65 year old subgroup, the follow-up period was short. Even though incidence rate in RA patients were high, short follow-up period may affect confidence interval of hazard ratio. In non-diabetes subgroup, the incidence number of ischemic stroke was small (N=7) and most cases occurred within short period of follow-up. These relative differences of follow-up period affected the confidence interval of hazard ratio. 

Q: The p value for hypertension in Table 3 should be in patients with hypertension, not without

Author’s Response:

I appreciate your comment and understand your concerns. The P-values of the Table showed the difference between each comparative subgroup. For example, P-value was compared between hypertensive and non-hypertensive subgroups. After your comment, we revised the arrangement of the P-value within Table 3 as follows. 

Table 3. Subgroup analyses between the seropositive RA and control groups

Variables Seropositive RA Control Hazard Ratio

(95% CI) 

P-values

for difference

 N Incidence

Rate

(1000 person-

year) N Incidence

Rate

(1000 person-

year) 

Sex Male 18 4.80 178 4.65 1.12 (0.682-1.835) 

P<0.001

 Female 46 4.07 334 2.95 1.44 (1.048-1.971) 

Age <65 38 2.81 333 2.49 1.30 (0.922-1.832) 

P<0.001

 ≥65 26 14.33 179 1.02 1.37 (0.900-2.082) 

Diabetes N 7 0.49 100 0.73 1.47 (1.106-1.952) 

P<0.001

 Y 57 59.60 412 28.14 0.98 (0.452-2.124) 

Hypertension N 29 2.72 240 2.48 1.14 (0.769-1.687) 

P<0.001

 Y 35 7.54 272 4.99 1.66 (1.156-2.378) 

Dyslipidemia N 55 4.27 403 3.24 1.43 (1.073-1.912) 

P=0.008

 Y 9 3.69 109 4.02 0.92 (0.462-1.833) 

Abbreviations: RA, rheumatoid arthritis; CI, confidence interval

Q: I would suggest deleting this statement “The hazard ratio of ischemic stroke among male patients in the seropositive RA group was 1.12 (95% CI, 0.68–1.84)”, since it was not significant and is shown in the Table.

Author’s Response:

I appreciate your comment. After your comment, we deleted the sentence. 

Discussion

Q: The discussion is somewhat difficult to follow; it lacks cohesion and fails to give potential explanations and/or show the relevance of some of the study findings.

Author’s Response:

I appreciate your opinion. After your comment, we revised the discussion part. We added a perspective about our findings on diabetes and hyperlipidemia. We mentioned the merits of our results compared to existing papers. The explanation about the mechanism of myocardial infarction with rheumatic disease was deleted. We corrected grammatical errors as well. 

Page 11 line 197

We analyzed the association between seropositive RA and ischemic stroke in Korea. Our nationwide longitudinal cohort study showed an increased risk of ischemic stroke in patients with seropositive RA. Furthermore, we conducted a subgroup analysis.

Solomon et al. [11] reported that the adjusted relative risk of myocardial infarction, not ischemic stroke, was higher in women with RA than that in women without RA. Similarly, Turesson et al. [15] reported that there was a trend toward an increased incidence of first episodes of ischemic stroke in RA cohort; however, the difference was not statistically significant. In a nationwide Swedish rheumatology cohort, the hazard ratio for ischemic stroke was 1.29 (95% CI, 1.18–1.41) in RA patients [10]. These findings are similar to ours; in our study, the age- and sex-adjusted (model 1) hazard ratio of ischemic stroke in the seropositive RA group was 1.32 (95% CI, 1.02–1.73). In model 2 (which adjusted for comorbidities in addition to age and sex), the incidence rate of ischemic stroke was 1.40 (95% CI, 1.07–1.82).

Several studies have found no association between ischemic stroke and RA [11,36,37]. These results suggest a lack of power due to an insufficient number of ischemic stroke events or a weaker association between RA and ischemic stroke. Our nationwide longitudinal cohort study helped to address some of the above uncertainties.

According to the subgroup analysis, in RA patients with hypertension, the hazard ratios of ischemic stroke risks was 1.66 (95% CI, 1.16-2.38). Our results suggest that the risk of ischemic stroke increases in hypertensive RA patients. Hypertension may have caused damaging effects in the blood vessels of these patients. Accelerated atherosclerosis caused by inflammation may have increased arterial stiffness [41], which may explain the evolution of ischemic stroke risks observed in our cohort.

 Our results showed that the prevalence of diabetes and hyprelipidemia was lower in the RA group than in the control group. The prevalence of diabetes was lower in the RA group (6.26%) than in the control group (10.10%) (p <0.001). The prevalence of hyperlipidemia was lower in the RA group (15.55%) than in the control group (17.84%) (p = 0.004). The above-mentioned results are similar to those reported in other studies [41,48]. One study [41] reported that diabetes mellitus tended to be less common in the RA group than in controls, and that the RA patients (4.6−6.3 mmol/L) had significantly lower total serum cholesterol than that in controls (5.4−7.0 mmol/L) (P < 0.001). A previous study [48] supported our results by reporting a lesser prevalence of hyperlipidemia in the RA group. In some studies [49,50], diabetes showed a higher prevalence in the RA group than in the control group, but there was no significant difference in the prevalence of hyperlipidemia between the two groups. There results demonstrates controversy in the prevalence of diabetes and hyperlipidemia among patients with RA compared to controls. Therefore, further studies may be warranted.

 Smoking is a well-established risk factor for RA [38]. Moreover, smoking and inflammatory activity have been demonstrated to increase fibrinogen levels, and this increase is a known risk factor for thromboembolic events such as ischemic stroke [39]. Similar to smoking, hypercoagulation and thrombosis caused by the inflammatory reaction in RA, which leads to embolization, are described as factors that increase the risk of ischemic stroke in RA.

Page 14 line 256

Nevertheless, to the best of our knowledge, this is the first study conducted in Korea that evaluates the association between ischemic stroke and seropositive RA in Korean patients. Our study is a nationwide, longitudinal study which included a large national data set based on health check-ups, with a follow-up that provided unified data to overcome the limitations of existing research results that used small amounts of data [41,48,49,50]. A large number of RA patients and their age-sex matched controls in a ratio of 1:5 also helped overcome the limitations of very low ischemic stroke incidence.

Q: As I mentioned in my previous comments, the different prevalence of diabetes and dyslipidemia between both groups should be added to the discussion (not just mentioned in the Results section).

Author’s Response:

I appreciate you for these helpful suggestions. After your comment, the different prevalence of diabetes and dyslipidemia between both groups were added to the discussion as follows.

Page 12, line 220

Our results showed that the prevalence of diabetes and hyprelipidemia was lower in the RA group than in the control group. The prevalence of diabetes was lower in the RA group (6.26%) than in the control group (10.10%) (p <0.001). The prevalence of hyperlipidemia was lower in the RA group (15.55%) than in the control group (17.84%) (p = 0.004). The above-mentioned results are similar to those reported in other studies [41,48]. One study [41] reported that diabetes mellitus tended to be less common in the RA group than in controls, and that the RA patients (4.6−6.3 mmol/L) had significantly lower total serum cholesterol than that in controls (5.4−7.0 mmol/L) (P < 0.001). A previous study [48] supported our results by reporting a lesser prevalence of hyperlipidemia in the RA group. In some studies [49,50], diabetes showed a higher prevalence in the RA group than in the control group, but there was no significant difference in the prevalence of hyperlipidemia between the two groups. There results demonstrates controversy in the prevalence of diabetes and hyperlipidemia among patients with RA compared to controls. Therefore, further studies may be warranted.

Q: The paragraph in lines 232 to 236 is not clear. The differential outcomes of myocardial infarction and stroke and then the different mechanisms leading to both of them do not appear to be related.

Author’s Response:

I appreciate your comment and agree with you. After your comment, we deleted those sentences.

---

## [Decision Letter · Decision Letter 2]

5 May 2021

Association between ischemic stroke and seropositive rheumatoid arthritis in Korea: A nationwide longitudinal cohort study

PONE-D-20-30296R2

Dear Dr. Sohn,

We’re pleased to inform you that your manuscript has been judged scientifically suitable for publication and will be formally accepted for publication once it meets all outstanding technical requirements.

Kind regards,

Miguel A. Barboza, MD, MSc

Academic Editor

PLOS ONE

Additional Editor Comments (optional):

Reviewers' comments:

Reviewer's Responses to Questions

**Comments to the Author**

1. If the authors have adequately addressed your comments raised in a previous round of review and you feel that this manuscript is now acceptable for publication, you may indicate that here to bypass the “Comments to the Author” section, enter your conflict of interest statement in the “Confidential to Editor” section, and submit your "Accept" recommendation.

Reviewer #2: (No Response)

2. Is the manuscript technically sound, and do the data support the conclusions?

Reviewer #2: Partly

3. Has the statistical analysis been performed appropriately and rigorously? 

Reviewer #2: Yes

4. Have the authors made all data underlying the findings in their manuscript fully available?

Reviewer #2: Yes

5. Is the manuscript presented in an intelligible fashion and written in standard English?

Reviewer #2: Yes

6. Review Comments to the Author

Reviewer #2: Most of my comments have been correctly addressed and I consider the manuscript has substantially improved. However, as I had already stated, I have one major concern regarding results depicted on Table 3.

According to those results, RA patients without diabetes have a lower incidence rate of stroke than non-diabetic controls (0.49 in RA vs 0.73 in non-RA controls), but the HR suggests RA is a risk factor for stroke in this group of patients (1.47; 1.106-1.952). On the other hand, RA patient with diabetes have a higher incidence rate than patients with diabetes and without RA, but the HR shows RA as a protective factor (although non-significant). Are those incidence rates correct?

7. PLOS authors have the option to publish the peer review history of their article (what does this mean?). If published, this will include your full peer review and any attached files.

Reviewer #2: No

---

## [Editor Report · Acceptance letter]

7 May 2021

PONE-D-20-30296R2 

Association between ischemic stroke and seropositive rheumatoid arthritis in Korea: A nationwide longitudinal cohort study 

Dear Dr. Sohn:

I'm pleased to inform you that your manuscript has been deemed suitable for publication in PLOS ONE. Congratulations! Your manuscript is now with our production department. 

Kind regards, 

on behalf of

Dr. Miguel A. Barboza 

Academic Editor

PLOS ONE